# VARIATIONAL CLASSIFICATION

## ABSTRACT

Classification tasks, ubiquitous across machine learning, are commonly tackled by
a suitably designed neural network with a *softmax* output layer, mapping each data
point to a categorical distribution over class labels. We extend this familiar model
from a latent variable perspective to *variational classification* (VC), analogous to
how the variational auto-encoder relates to its deterministic counterpart. We derive
a training objective based on the ELBO together with an *adversarial* approach
for optimising it. Within this framework, we identify design choices made im-
plicitly in off-the-shelf softmax functions and can instead include domain-specific
assumptions, such as class-conditional latent priors. We demonstrate benefits of
the VC model in image classification. We show on several standard datasets, that
treating inputs to the softmax layer as latent variables under a mixture of Gaussians
prior, improves several desirable aspects of a classifier, such as prediction accuracy,
calibration, out of domain calibration and adversarial robustness.

## 1 INTRODUCTION

Classification is central to much of machine learning, not only in its own right, e.g. to categorise every
day objects (Klasson et al., 2019), make medical diagnoses (Adem et al., 2019; Mirbabaie et al., 2021)
or detect potentially life-supporting planets (Tiensuu et al., 2019), but also as an important component
in other learning paradigms, e.g. to select actions in reinforcement learning, distinguish positive
and negative samples in contrastive learning or within the attention mechanism of large language
models. Recently, it has become all but default to tackle classification tasks with domain-specific
neural networks with a *sigmoid* or *softmax* output layer.[1] The neural network deterministically maps
each data point $x$ (in a domain $\mathcal{X}$) to a real vector $f_\omega(x)$, which the last layer maps to the parameter
of a discrete distribution $p_\theta(y|x)$ over class labels $y \in \mathcal{Y}$, defined by a point on the simplex $\Delta^{|\mathcal{Y}|}$, e.g.:

$$p_\theta(y|x) = \text{softmax}(x;\theta)_y = \frac{\exp g(x,y;\theta)}{\sum_{y' \in \mathcal{Y}} \exp g(x,y';\theta)} = \frac{\exp\{f_\omega(x)^\top w_y + b_y\}}{\sum_{y' \in \mathcal{Y}} \exp\{f_\omega(x)^\top w_{y'} + b_{y'}\}} \quad (1)$$

Despite frequently outperforming alternatives and their widespread use, softmax classifiers are
not without issue. The overall mapping from $\mathcal{X}$ to $\Delta^{|\mathcal{Y}|}$ is learned numerically by minimising
a loss function over a finite set of training samples. The result is poorly understood in general,
remaining in many respects a "black box" with predictions hard to rationalise. A trained classifier
may make accurate predictions for the training set, but predictions for other data points, e.g. test
data, are determined by $f_\omega \in \mathcal{F}$, from a class of functions chosen to be highly flexible in the hope of
approximating the unknown true mapping $f(x) = \{p(y|x)\}_{y \in \mathcal{Y}}$. With sufficient flexibility, (i) $\mathcal{F}$ may
contain many, possibly infinite, functions (for different $\omega$) that give accurate training set predictions,
but dissimilar and hence uncertain predictions elsewhere; and (ii) predictions can change materially
for imperceptible changes in the data (*adversarial examples*). Lastly, where $p_\theta(y|x)$ fails to reflect
the true label distribution $p(y|x)$, it fails to reflect the frequency with which classes are expected to
occur, making the classifier *miscalibrated*.

A standard softmax classifier also lacks several desirable properties. For example, under certain
conditions, it is known that a softmax classifier learns to approximate $p(y|x)$ and so captures
stochastic, or *aleatoric*, uncertainty in the data. However, in practice, predictions for some regions of
$\mathcal{X}$ may be more reliable than others and it can be important to understand the confidence or *epistemic*

---

[1]Since the softmax function generalises the sigmoid function to more than two classes, we refer to softmax
throughout, but all arguments can be applied to the sigmoid case.

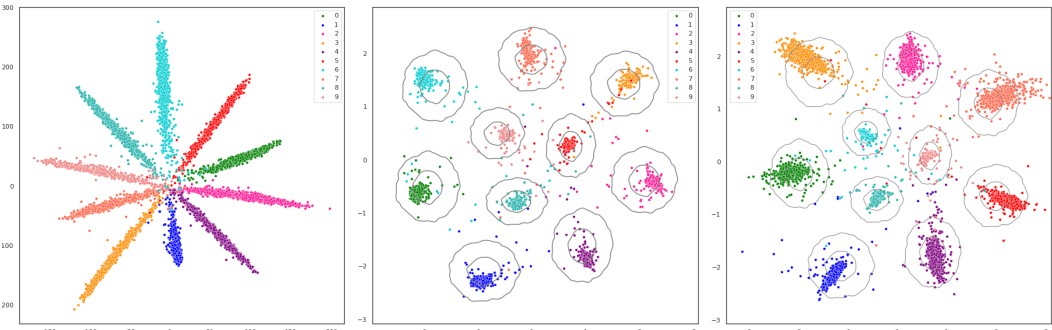

Figure 1: Distributions of softmax inputs $q_\phi(z|y)$, $z \in \mathbb{R}^2$, under the three VC training objectives (MNIST dataset): (*left*) standard softmax Cross Entropy - "MLE" treatment; (*centre*) with class-conditional Gaussian priors $p_\theta(z|y)$ - "MAP" treatment; (*right*) and entropy - "Bayesian" treatment. (Note: softmax inputs have been artificially restricted to 2-dimensional for visualisation purposes.)

uncertainty of predictions. Also, beyond making positive classifications from a set of labels, it can be useful to know when a sample is "none of the above", or *out of distribution*. In such cases, a softmax might output a uniform distribution over all classes but that is equally appropriate for a fully *in-distribution* sample that occurs with all labels, and so is ambiguous without further assumptions.

Overall, classification accuracy, model calibration and adversarial robustness depend on how well a model $p_\theta(y|x)$ approximates the true distribution $p(y|x)$, in particular how it interpolates/extrapolates from the training set to $\mathcal{X}$. Meanwhile, prediction confidence and out-of-sample detection depend on how new data samples compare to those seen at training time. This suggests that to improve a softmax classifier in general requires (i) modelling $p(y|x)$ more accurately, e.g. by obtaining more data or constraining $f_\omega$ in a useful, perhaps domain-specific way; and (ii) developing a measure of *familiarity* between data samples, e.g. by learning a distribution over $x$, or similar.

We approach this by generalising the mechanics of the softmax classifier guided by two key observations: (i) the parallel between softmax $p_\theta(y|x) = \frac{exp\{g(x,y)\}}{\sum_\mathcal{Y} exp\{g(x,y')\}}$ and Bayes' rule $p(y|x) = \frac{p(x,y)}{\sum_\mathcal{Y} p(x,y')}$; and (ii) that each layer of a neural network classifier is a function of a random variable (the data x) and so can be treated as a *latent* random variable z. The first hints at a probabilistic interpretation of the softmax function. The second also underpins the relationship between the *deterministic auto-encoder* and the *variational auto-encoder* (VAE) (Kingma & Welling, 2014; Rezende et al., 2014), the latter of which both generalises and constrains the former. In a similar way, we propose the *variational classifier* (VC) that introduces a latent variable z into the (Markov) prediction model $p_\theta(y|x) = \int_z p_\theta(y|z)p_\theta(z|x)$, with marginal $p(z)$. This offers a promising, principled way to address the issues outlined if $p(z)$ both usefully constrains the model to improve accuracy; and, when evaluated for latent variables associated with observed data, indicates their "familiarity". We develop an analog of the ELBO to train a VC and propose an "adversarial trick" for its optimisation.

We show that the standard softmax classifier falls within the VC framework under distributional assumptions that equate to implicit design choices. By identifying such choices, alternatives can be introduced where appropriate, such as domain-specific latent priors. This also shows that the VC framework does not "complicate matters" by requiring difficult distribution choices to be made, rather it exposes that *default assumptions are made in softmax classifiers* that may not be optimal. A regular softmax classifier is also seen to concentrate class-conditional latent distributions $q_\phi(z|y) = \int_x q_\phi(z|x)p(x|y)$, to a single point, akin to a *maximum likelihood* point estimate, whereas a VC fits $q_\phi(z|y)$ to a class-conditional prior $p_\theta(z|y)$, mirroring a more *Bayesian* treatment (see Figure 1).

On a series of image classification experiments, we demonstrate that a VC outperforms a regular softmax classifier in many of the ways outlined, such as calibration, including for out of domain samples, adversarial robustness, and modestly improves accuracy, more notably if data is scarce.

We believe the VC framework offers a deeper interpretation of softmax classification and takes a step towards more fully understanding these familiar models, potentially enabling their further improvement and/or integration with other latent variable models, e.g. VAEs or contrastive learning.

## 2 BACKGROUND (THE VARIATIONAL AUTO-ENCODER)

The proposed generalisation from softmax to variational classification is highly analogous to how the deterministic auto-encoder relates to the variational auto-encoder, which we briefly summarise.

Maximising the likelihood of the data to estimate parameters of the latent variable model $p_\theta(\mathrm{x}) = \int_z p_\theta(\mathrm{x}|\mathrm{z})p_\theta(\mathrm{z})$ is intractable in general, so the *evidence lower bound* (ELBO) is maximised instead:

$$
\int_x p(x) \log p_\theta(x) = \int_x p(x) \int_z q_\phi(z|x) \Big\{ \log p_\theta(x|z) - \log \tfrac{q_\phi(z|x)}{p_\theta(z)} + \log \tfrac{q_\phi(z|x)}{p_\theta(z|x)} \Big\}
$$

$$
\geq \int_x p(x) \Big\{ \int_z q_\phi(z|x) \log p_\theta(x|z) - \int_z q_\phi(z|x) \log \tfrac{q_\phi(z|x)}{p_\theta(z)} \Big\} \doteq \text{ELBO} \quad (2)
$$

Maximising the ELBO in equation 2 is equivalent to minimising

$$
D_{\text{KL}}\big[\, p(x) \,\|\, p_\theta(x)\,\big] \;+\; \mathbb{E}_x\big[D_{\text{KL}}\big[\, q_\phi(z|x) \,\|\, p_\theta(z|x)\,\big]\big], \tag{3}
$$

where $D_{\text{KL}}\big[\, p(x) \,\|\, q(x)\,\big] = \int_x p(x) \log \tfrac{p(x)}{q(x)}$ is the Kullback-Leibler (KL) divergence. This shows that maximising the ELBO fits the model $p_\theta(x)$ to the data distribution $p(x)$, whilst fitting the approximate posterior $q_\phi(z|x)$ to the implied posterior under the model $p_\theta(z|x) = \tfrac{p_\theta(x|z)p_\theta(z)}{p_\theta(x)}$; or ensuring that the two modelled distributions $q_\phi(z|x)$ and $p_\theta(x|z)$ are *consistent under Bayes' rule*.

The *Variational Auto-Encoder* (VAE) (Kingma & Welling, 2014; Rezende et al., 2014) implements the ELBO with the parameters of $p_\theta$ and $q_\phi$ defined by flexible neural networks. Restricting the variance of $q_\phi$ towards zero such that $q_\phi$ tends to a delta distribution, the first term of equation 2 tends to the loss function of a deterministic auto-encoder. As such, the VAE can be seen to generalise the deterministic case, allowing for uncertainty or stochasticity in the latent variables while constraining their marginal distribution $p(\mathrm{z})$ with the second "regularisation" term.

## 3 VARIATIONAL CLASSIFICATION

We now present the the (latent) variational classifier (**VC**) as a generalisation of the softmax classifier.

A standard softmax classifier is a deterministic function that maps each data point $x \in \mathcal{X}$ through a sequence of intermediate representations to a *prediction* $p_\theta(y|x)$, a categorical distribution over labels $y \in \mathcal{Y}$, defined by a point on the simplex $\Delta^{|\mathcal{Y}|}$. Treating each data sample $x$ as a realisation of a random variable x with distribution $p(\mathrm{x})$, any intermediate representation $z = g(x)$ can be considered the realisation of a latent random variable z with distribution $p(\mathrm{z})$ defined implicitly by sampling $x \sim p(\mathrm{x})$. Assuming the Markov property $\underline{\mathrm{x} \rightarrow \mathrm{z} \rightarrow \mathrm{y}}$, the latent **VC model** is given by:

$$
p_\theta(y|x) = \int_z p_\theta(y|z) q_\phi(z|x) \tag{4}
$$

While equation 4 is more general, we keep softmax classification in mind with a running example where z is the input to the softmax layer; $q_\phi(z|x) = \delta_{z - f_\omega(x)}$, a delta distribution parameterised by the neural network up to the softmax layer $f_\omega(\cdot)$; and $p_\theta(\mathrm{y}|\mathrm{z})$ is a categorical label distribution defined by the softmax layer. We now derive an objective to learn parameters of the VC model.

Similarly to the latent variable model for $p_\theta(\mathrm{x})$ (section 2), the VC model cannot generally be learned by likelihood maximisation, hence we compute a lower bound comparable to the ELBO (eq. 2):

$$
\int_{x,y} p(x,y) \log p_\theta(y|x) = \int_{x,y} p(x,y) \Big\{ \int_z q_\phi(z|x) \log p_\theta(y|z,x) + \int_z q_\phi(z|x) \log \tfrac{q_\phi(z|x)}{p_\theta(z|x,y)} \Big\}
$$

$$
\geq \int_{x,y} p(x,y) \int_z q_\phi(z|x) \log p_\theta(y|z,x) = \int_{x,y} p(x,y) \int_z q_\phi(z|x) \log p_\theta(y|z) , \tag{5}
$$

where $p_\theta(y|x,z) = p_\theta(y|z)$ in the last step (by Markov). A strict analogue of the ELBO would replace $x$ by $y$ in equation 2 and condition throughout on a (new) $x$, such that the variational distribution $q_\phi$ depends on both $x$ and $y$ (e.g. see Tang & Salakhutdinov, 2013). Instead, $q_\phi$ is chosen to depend only on $x$, hence the KL term $D_{\text{KL}}\big[\, q_\phi(z|x) \,\|\, p_\theta(z|x,y)\,\big]$ (dropped in line 1) is minimised but, in

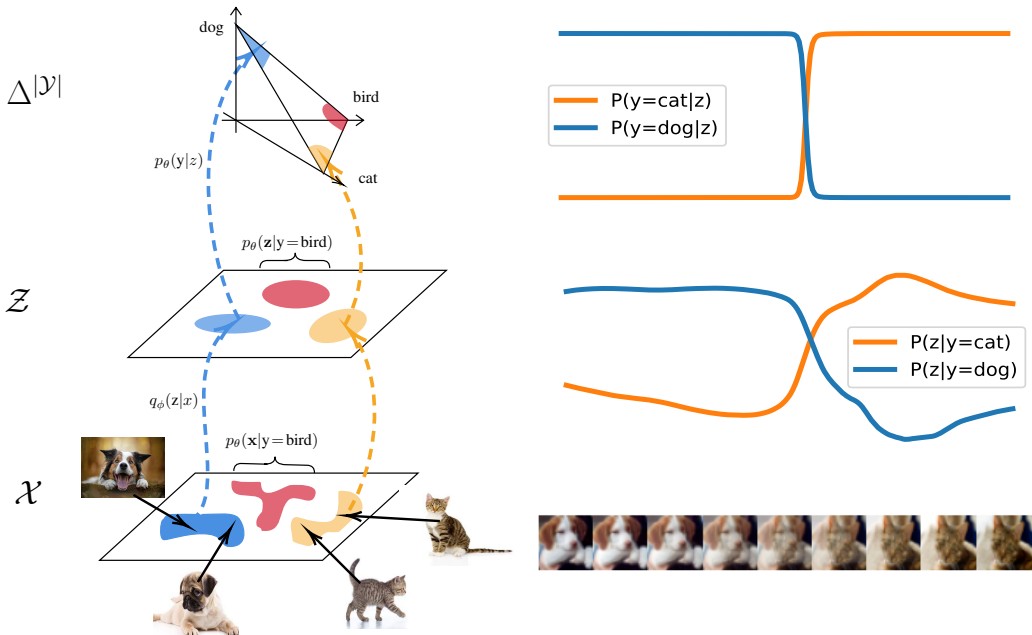

Figure 2: The Variation Classier: *(l)* $q_\phi(z|x)$ stochastically maps data to the latent domain, where class-conditional distributions $q_\phi(z|y)$ are fitted to priors $p_\theta(z|y)$ with known analytical form; class predictions $p_\theta(y|z)$ are "read off" by Bayes' rule. *(r-bottom)* images from a line in the data domain $\mathcal{X}$ that passes through two data samples (dog and cat images); *(r-middle)* $p(z)$ for the corresponding latent variables $z = f_\omega(x)$; *(r-top)* corresponding output predictions $p_\theta(y|x)$. Note that output predictions may seem confident even where $p(z)$ indicates lower confidence should be attributed.

general, may not vanish when maximising the lower bound (final expression). However, we do have $D_{\mathrm{KL}}[\,q_\phi(z|x)\|\,p_\theta(z|x,y)] = 0$ in the limit that z is a *sufficient statistic* for y given x, i.e. z contains all information about y in x.[2] Hence maximising the lower bound and so minimising $D_{\mathrm{KL}}[\,q_\phi(z|x)\|\,p_\theta(z|x,y)]$ pushes z towards a sufficient statistic.

So far, the training objective (last term, eq. 5) closely resembles cross-entropy, which it reduces to if $q_\phi(z|x)$ is set to a delta function. However, now, the modelled distribution $p_\theta(y|z)$ and the implied distribution $q_\phi(z|y) = \int_x q_\phi(z|x)p(x|y)$ must be *consistent under Bayes' rule*, analogous to $p_\theta(x|z)$ and $q_\phi(z|x)$ in the ELBO (section 2). Hence, by modelling $p_\theta(y|z) = \frac{p_\theta(z|y)p_\theta(y)}{\sum_{y'} p_\theta(z|y')p_\theta(y')}$, we should have $p_\theta(z|y) = q_\phi(z|y)$, or $D_{\mathrm{KL}}[\,q_\phi(z|y)\|\,p_\theta(z|y)] = 0$.[3] To clarify, in the general case, for a class $y$: $q_\phi(z|y)$ is an implicit distribution defined by putting samples $x \sim p(x|y)$ through the neural network parameterising $q_\phi(z|x)$ and sampling therefrom; whereas $p_\theta(z|y)$ is a class-conditional latent prior defined analytically. In the running softmax example, $q_\phi(z|y)$ is given directly by putting samples from class $y$ through the neural network: $z = f_\omega(x), x \sim p(x|y)$.

Including the KL constraint and a model of the class distribution $p_\pi(y)$, we maximise (**VC objective**):

$$\int_{x,y} p(x,y)\left\{ \int_z q_\phi(z|x) \log p_\theta(y|z) - \int_z q_\phi(z|y) \log \frac{q_\phi(z|y)}{p_\theta(z|y)} + \log p_\pi(y) \right\} \qquad (6)$$

Taken incrementally, terms of the VC objective involving $q_\phi$ (note the KL term has 2) can be seen to treat the latent z from a *maximum likelihood*, *maximum a posteriori* and *Bayesian* perspective:

(i)  maximising $\int_z q_\phi(z|x) \log p_\theta(y|z)$ gives $q_\phi(z|x) = \delta_{z-z_x}$, $z_x = \arg\max_z p_\theta(y|z)$     [MLE]

(ii) maximising the *prior* $\int_z q_\phi(z|y) \log p_\theta(z|y)$ alters the above point estimate $z_x$     [MAP]

(iii) maximising *entropy* $-\int_z q_\phi(z|y) \log q_\phi(z|y)$ has $q_\phi(z|y)$ "fill" the prior $p_\theta(z|y)$     [Bayesian]

---

[2] $p(z|x,y)p(y|x) = p(y|x,z)p(z|x) \implies p(z|x,y) = p(z|x) \Leftrightarrow p(y|x) = p(y|x,z) = p(y|z)$ (by Markov)
[3] Note that the analogous term in the ELBO, $D_{\mathrm{KL}}[\,q_\phi(z|x)\|\,p_\theta(z|x)]$ (conditioned on $x$ rather than $y$), is minimised implicitly when maximising the ELBO, whereas we minimise $D_{\mathrm{KL}}[\,q_\phi(z|y)\|\,p_\theta(z|y)]$ explicitly.

Comparing to the KL form of the ELBO (equation 3), maximising the VC objective minimises:

$$\mathbb{E}_x\big[D_{\mathrm{KL}}[\,p(y|x)\|\,p_\theta(y|x)] + \mathbb{E}_{x,y}\big[D_{\mathrm{KL}}[\,q_\phi(z|x)\|\,p_\theta(z|x,y)]\big]$$
$$+ \mathbb{E}_y\big[D_{\mathrm{KL}}[\,q_\phi(z|y)\|\,p_\theta(z|y)]\big] + D_{\mathrm{KL}}[\,p(y)\|\,p_\pi(y)]\big]\,, \quad (7)$$

revealing how the central objective of modelling $p(y|x)$ by $p_\theta(y|x)$ is constrained. Figure 2 gives an overview of the VC model. Components of a variational classifier can be interpreted as follows:

- the neural network up to the (generalised) softmax layer ($f_\omega$) effectively transform a mixture of analytically unknown class-conditional distributions $p(\mathrm{x}|y)$ over $\mathcal{X}$, to a mixture of analytically defined distributions $p_\theta(\mathrm{z}|y)$ over $\mathcal{Z}$;

- with inputs following an anticipated mixture of class-conditional distributions, the (generalised) softmax layer effectively "reads off" class predictions by Bayes' rule.

## 3.1 RELATIONSHIP TO SOFTMAX

Training a Variational Classifier by maximising its objective (eq. 6) requires $q_\phi(\mathrm{z}|\mathrm{x})$ and $p_\theta(\mathrm{z}|\mathrm{y})$ to be defined ($p_\pi(\mathrm{y})$ is assumed categorical for classification). While the VC objective applies more generally, as in the running example we now treat softmax layer inputs as samples of z and $q_\phi(\mathrm{z}|x)$ a delta function parameterised by the neural network: $z|x = f_\omega(x)$; $q_\phi(z|x) = \delta_{z-f_\omega(x)}$. Under these assumptions, the first term of the VC objective equates to softmax cross-entropy (SCE)

$$\int_z q_\phi(z|x) \log \frac{p_\theta(z|y)p_\theta(y)}{\sum_{y'\in\mathcal{Y}} p_\theta(z|y')p_\theta(y')} = p_\theta(y|x) = \log \frac{\exp\{f_\omega(x)^\top w_y + b_y\}}{\sum_{y'\in\mathcal{Y}} \exp\{f_\omega(x)^\top w_{y'} + b_{y'}\}}\,, \quad (8)$$

if $p_\theta(z|y) = \exp\{z^\top w_y + b'_y + c(z)\}$, $\forall y \in \mathcal{Y}$, where $c(z)$ may depend on $z$ but not $y$, and $b'_y$ absorbs $p_\theta(y)$. This shows that the standard softmax cross-entropy loss fits the "MLE" form of the VC objective (i.e. the first term only), under the implicit assumptions $q_\phi(\mathrm{z}|x)$ is a delta distribution ($\forall x$), and latent variables follow a mixture of (similar) class-conditional *exponential family distributions*, e.g. equivariate Gaussians. We note a close analogy to how a deterministic auto-encoder fits within the variational auto-encoder framework.

In corresponding to the MLE form of the VC objective, softmax cross-entropy (SCE) loss *accommodates* the softmax input distribution $q_\phi(\mathrm{z}|y)$ (i.e. $z = f_\omega(x)$, $x \sim p(\mathrm{x}|y)$) following the class prior $p_\theta(\mathrm{z}|y)$, but *does not encourage samples to fit it*. Indeed, SCE is optimised if $q_\phi(\mathrm{z}|y)$ "collapses" to a single point maximising $p_\theta(y|z)$ (see (i), Section 3). Figure 1 shows the softmax input distributions $q_\phi(\mathrm{z}|y)$ as each term of the VC objective is included: the prior has a clear impact (*left* to *centre*), whereas the entropy effect is more subtle (*centre* to *right*). We also see that, in practice, $q_\phi(\mathrm{z}|y)$ for each class $y$ does not fully collapse to a point, likely due to constraints on $f_\omega$, e.g. $\ell_2$ regularisation.

Since softmax classification is a special case of the VC framework, the latter does not add unnecessary complexity by requiring difficult distributional assumptions to be made, rather it exposes that *unscrutinised assumptions lie within softmax classifiers*. By generalising the softmax case, the variational classifier enables such assumptions to be varied, e.g. to use domain-specific priors $p_\theta(\mathrm{z}|y)$, and extends the MLE latent variable model to a fuller Bayesian-like treatment.

A softmax classifier is readily extended to a VC by: (i) optimising the full VC objective (eq. 6) such that softmax inputs for a class $y$, i.e. $z \sim q_\phi(\mathrm{z}|y)$, are encouraged to fit $p_\theta(\mathrm{z}|y)$; and (ii) relaxing the constraint that class-conditional priors are similar (e.g. equivariant) exponential family distributions, e.g. letting $p_\theta(\mathrm{z}|y)$ for each class $y$ be a multivariate Gaussian with parameters learned from the data.

## 3.2 OPTIMISING THE VC OBJECTIVE

Of the VC objective (eq. 6):

- the first term can be calculated by sampling $q_\phi(\mathrm{z}|x)$ (using the "reparameterisation trick" if necessary (Kingma & Welling, 2014)) and computing $p_\theta(\mathrm{y}|\mathrm{z})$ by Bayes' rule;

- the third term standard multinomial cross-entropy; but

- the second term is less immediate since $q_\phi(\mathrm{z}|y)$ is an implicit distribution, which cannot be evaluated only sampled, by sampling $z \sim q_\phi(\mathrm{z}|x)$, defined by the neural network, for $x \sim p(\mathrm{x}|y)$.

Fortunately, we only require *log ratios*, $\log \frac{q_\phi(z|y)}{p_\theta(z|y)}$, $\forall y \in \mathcal{Y}$, which can each be approximated by training a binary classifier to distinguish samples of $q_\phi(\text{z}|y)$ from $p_\theta(\text{z}|y)$. This *contrastive* "trick" has become increasingly common and underpins learning methods such as Noise Contrastive Estimation (Gutmann & Hyvärinen, 2010) and Contrastive Learning (Oord et al., 2018) and has been used in a similar way to train variants of the VAE (Makhzani et al., 2015; Mescheder et al., 2017).

Specifically, we maximise an *auxiliary objective* w.r.t. the parameters $\psi$ of a set of binary classifiers:

$$\int_y p(y)\Big\{ \int_z q_\phi(z|y) \log \sigma(T^y_\psi(z)) \ + \int_z p_\theta(z|y) \log(1 - \sigma(T^y_\psi(z))) \Big\} \tag{9}$$

where $\sigma$ is the logistic sigmoid function $\sigma(x) = (1 + e^{-x})^{-1}$, $T^y_\psi(z) = w_y^\top z + b_y$ and $\psi = \{w_y, b_y\}_y$. It is easy to show that equation 9 is optimised if $T^y_\psi(z) = \log \frac{q_\phi(z|y)}{p_\theta(z|y)}$, $\forall y \in \mathcal{Y}$, hence when all binary classifiers are trained, $\{T^y_\psi(z)\}_{y \in \mathcal{Y}}$ approximate the log ratios required by the VC objective.

This approach is *adversarial* since the VC objective is maximised when the log ratios are *minimsed*, i.e. $q_\phi(z|y) = p_\theta(z|y)$ and samples produced by the neural network are indistinguishable from those of the prior; whereas the auxiliary objective is maximised when the log ratio are *maximised* and the two distributions discriminated. Any distributional differences that the auxiliary binary classifiers identify are removed under the main objective by bringing the distributions closer together, until the distributions match. Similar to Generative Adversarial Networks (GANs) (Goodfellow et al., 2014), the neural network $f_\omega$ can be considered a *generator* and each binary classifier a *discriminator*, however, we require one discriminator per class that each distinguish generated samples from a *learned* rather than fixed reference distribution; and whereas a GAN discriminator typically distinguishes between highly complex distributions, each VC discriminator simply compares a Gaussian to an approximate Gaussian, requiring only logistic regression.

In principle, we require gradients of the log ratios w.r.t parameters $\theta$, $\phi$ of the VC objective. However, the gradient w.r.t. the occurrence of $\phi$ in the log ratio is zero (Mescheder et al., 2017) and that w.r.t. $\theta$ can be computed directly from equation 6 and no gradients from the binary classifiers are required.

### 3.3 INTERPRETING VARIATIONAL CLASSIFICATION

Variational classification might be interpreted in one of two ways:

**Well-specified generative model**: Assume data $x \in \mathcal{X}$ is generated according to a hierarchical model: y $\to$ z $\to$ x, where $p(\text{y})$ is categorical; $p(\text{z}|y)$ are analytically known distributions with z of manageable dimension, e.g. $\mathcal{N}(\mu_y, \Sigma_y)$; and $x = h(z)$ for an arbitrary invertible function $h : \mathcal{Z} \to \mathcal{X}$ (if $\mathcal{X}$ is of higher dimension than $\mathcal{Z}$, assume $h$ maps one-to-one to a manifold in $\mathcal{X}$). Hence $p(\text{x})$ is a mixture of unknown distributions. By choosing $p_\theta(\text{z}|y)$ to have the form of the true $p(\text{z}|y)$, variational classification effectively aims to invert $h$ and learn the parameters of the true generative model. In practice, the model parameters including $h^{-1}$ may only be identifiable up to certain equivalences, however the learned latent variables may reflect true latent variables and be semantically meaningful.

**Miss-specified model**: Assume data is generated as above, but with z of vast, possibly uncountable dimension with complex inter-dependencies, e.g. determining every blade of grass in a landscape or every hair on a cat. It is an impossible task to learn all such latent variables with a lower dimensional model, in general. As such, the VC latent space might learn a complex function of many true latent variables.

The former scenario appeals since the model may learn disentangled, semantically meaningful features of the data. However, that requires well-specified distributions and the number of true latent variables may make it impossible. For natural data with many latent variables, the second case seems more likely and choosing $p_\theta(\text{z}|y)$ to be Gaussian may be justified by the Central Limit Theorem.

## 4 RELATED WORK

Noting that the softmax denominator satisfies $\sum_{y \in \mathcal{Y}} \exp g(x, y) \propto p_\theta(x)$, Grathwohl et al. (2019) view the softmax classifier as an *energy-based* (i.e. un-normalised probabilistic) model. Generative aspects aside, our work is comparable in the sense of taking an abstracted view of softmax classification to improve aspects of it. However, the benefits they obtain, e.g. to calibration and adversarial

robustness, come at a cost to the core aim of accurate classification. Further, the MCMC step they require for normalisation reportedly slows and destabilises training, whereas we use tractable probability distributions with little overhead.

Several previous works adapt the standard ELBO, used to learn a model of $p(x)$, to a conditional analog for learning $p(y|x)$ (Tang & Salakhutdinov, 2013; Sohn et al., 2015). However, such works focus on generative scenarios rather than classification, e.g. $x$ being a face image and $y|x$ being the same face in a different pose determined by latent $z$; or $x$ being part of an image and $y|x$ its completion given latent content $z$. The *Gaussian stochastic neural network* (GSNN) model (Sohn et al., 2015) takes a further step towards our own by conditioning $q(z|x, y)$ only on $x$, however none of these models considers the class-level latent distributions $q(z|y)$ in variational classification.

Variational classification effectively subsumes many works that add a regularisation term to the softmax cross-entropy loss function, interpretable as a prior over latent variables in the "MAP" VC model. For example, several semi-supervised learning models can be interpreted as treating softmax *outputs* as latent variables, where the prior guides the outputs of unlabelled data (e.g. see Allen et al., 2020). More closely related to variational classification, several works can be seen to treat the softmax *input* as a latent variable with a regularisation term applied to encourage certain prior beliefs. For example, label predictions can be encouraged to be deterministic (all probability mass on a single class) by enforcing a *large margin* between clusters in the latent space (Liu et al., 2016; Wen et al., 2016; Wan et al., 2018; 2022).

## 5 EMPIRICAL VALIDATION

We now aim to show empirically that the latent structure imposed by the variational classifier objective improves various useful properties relative to the standard softmax classifier. In principle, the VC model is expected to be applicable wherever a softmax classifier is used, if useful distributional assumptions can be made. We choose the visual domain and demonstrate performance of the variational classifier across a range of tasks on familiar datasets.

For fair comparison, we make minimal changes to adapt a standard softmax classifier to a variational classifier. As described in section 3.1, we train on the VC training objective (equation 6) under the following assumptions: $q_\phi(z|x)$ is a delta distribution parameterised by a neural network $f_\omega : \mathcal{X} \to \mathcal{Z}$; class-conditional priors $p_\theta(z|y)$ are multi-variate Gaussians with parameters learned from the data (we use diagonal covariance for simplicity). To examine the effect of each component of the VC objective, we compare classifiers trained to maximise three objective functions (see section 3):

- $J_{CE} = \int_{x,y} p(x, y)\{\int_z q_\phi(z|x) \log p_\theta(y|z) + \log p_\pi(y)\}$.

  This is equivalent to standard softmax cross-entropy under the above assumptions and corresponds to the MLE form of the VC objective (section 2, (i)). We refer to this model as CE.

- $J_{GM} = J_{CE} + \int_{x,y} p(x, y) \int_z q_\phi(z|y) \log p_\theta(z|y)$

  This includes class-conditional priors and corresponds to the MAP form of the VC objective (section 2, (ii)). We refer to this model as GM.

- $J_{VC} = J_{GM} - \int_{x,y} p(x, y) \int_z q_\phi(z|y) \log q_\phi(z|y)$

  This includes entropy of the generated latent class-conditional distributions and corresponds to the Bayesian form of the VC objective (section 2, (iii)). We refer to this model as VC.

### 5.1 ACCURACY AND CALIBRATION

We first look to compare the classification accuracy and calibration of each model on two standard benchmarks (CIFAR-10 and CIFAR-100), across two standard ResNet model architectures

| | CIFAR-10 | | | CIFAR-100 | | |
| --- | --- | --- | --- | --- | --- | --- |
| | **CE** | **GM** | **VC** | **CE** | **GM** | **VC** |
| *WideResNet-28-10* | 96.11 | 95.02 | **96.27** | 80.21 | 79.53 | **80.44** |
| *ResNet-50* | **93.76** | 92.97 | 93.28 | 73.19 | **74.31** | 73.42 |

Table 1: Classification Accuracy (%)

|  | CIFAR-10 | | | CIFAR-100 | | |
|---|---|---|---|---|---|---|
|  | **CE** | **GM** | **VC** | **CE** | **GM** | **VC** |
| *WideResNet-28-10* | 3.10 | 3.51 | **2.06** | 10.23 | 19.58 | **4.81** |
| *ResNet-50* | 3.72 | 4.13 | **3.17** | 8.72 | 10.60 | **5.21** |

Table 2: Expected Calibration Error (%) (lower is better).

(*WideResNet-28-10* and *ResNet-50*) . Calibration is evaluated in terms of the *Expected Calibration Error* (ECE) (Guo et al., 2017) (see Appendix A).

Table 1 shows that the VC model achieves a slight performance improvement for the more powerful *WideResNet-28-10* model, while performing competitively in general. However, Table 2 shows that the VC model is by far the most calibrated. We note that this calibration is not performed *post hoc* and requires no external calibration set, unlike approaches such as Platt's scaling and temperature scaling (Platt et al., 1999; Guo et al., 2017). Reliability diagrams comparing the VC and CE models are plotted in Figure 3.

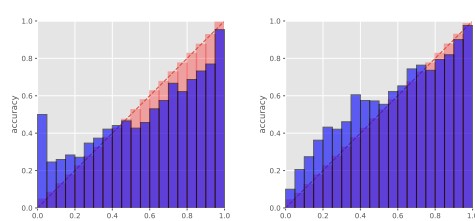

Figure 3: Reliability Diagrams: *(l)* CE, *(r)* VC

## 5.2 OUT OF DISTRIBUTION GENERALIZATION

We first test the ability to **detect OOD examples** by computing the AUROC when a model is trained on CIFAR-10 and evaluated on the validation set of CIFAR-10 mixed (in turn) with SVHN, CIFAR-100, and CELEBA (Goodfellow et al., 2013; Liu et al., 2015). We compare the VC and CE models using the probability of the predicted class $\arg\max_y p_\theta(y|x)$ as a means of identifying OOD samples.

| Model | SVHN | C-100 | CelebA |
|---|---|---|---|
| $P_{CE}(y|x)$ | 0.92 | 0.88 | 0.90 |
| $P_{VC}(y|z)$ | 0.93 | 0.86 | 0.89 |

Table 3: AUROC for the OOD detection task. Models are trained on CIFAR-10 and evaluated on in and out-of-domain samples.

Table 3 shows that VC does not perform better than the standard CE model. For the VC model, we also tried $p(z)$ as a metric to detect OOD samples and found it to perform comparably. Our results are fairly consistent with those of Grathwohl et al. (2019). We conclude that although the VC model learns to map training samples to a more structured latent space, this does not extend to OOD data, which are mapped randomly and hence we perform comparably to CE.

When deployed in real-world settings, machine learning models can encounter *distribution shift* relative to the training data. It can be imperative to know when the model output is reliable and can be trusted, requiring that models to be **calibrated on OOD data** and *know when they do not know*. To test performance under distribution shift, we look to the robustness benchmarks proposed by Hendrycks & Dietterich (2019); they simulate distribution shift by adding varying intensities of different corruptions to a dataset. We compare the CE to VC models on two proposed benchmarks: CIFAR-10-C and CIFAR-100-C.

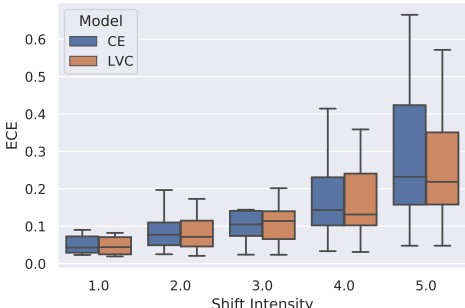 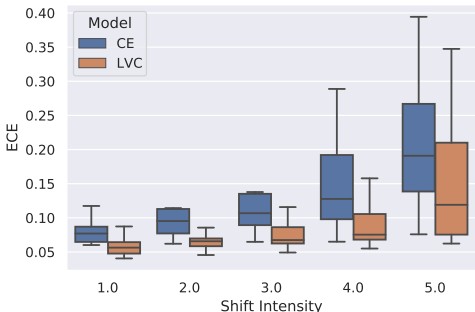

Figure 4: Calibration under *distribution shift* (corruption). *(l)* CIFAR-10-C, *(r)* CIFAR-100-C

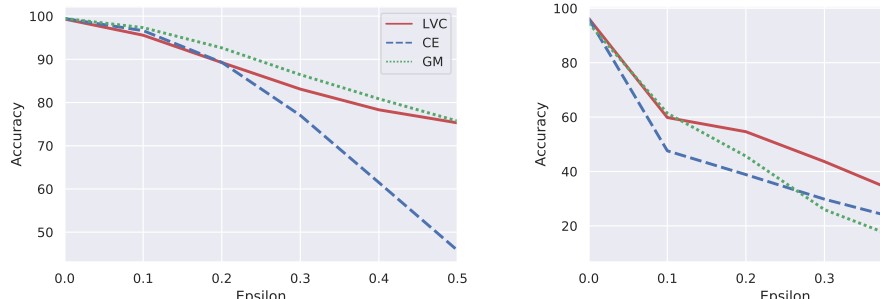

Figure 5: Prediction accuracy as FGSM adversarial attacks increase *(l)* MNIST; *(r)* CIFAR-10

Both models perform comparably in terms of classification accuracy (see Appendix), which accords with the results above. However, as shown in Figure 4, VC has a consistently lower calibration error as the corruption intensity increases (left to right). We note that the benefit of the VC model is more noticeable in the more challenging CIFAR-100-C task (right plot).

### 5.3 ADVERSARIAL ROBUSTNESS

Here, we look to understand if regularising a classifier's latent structure improves adversarial robustness. We test model robustness by measuting performance on adversarially generated images using the common *Fast Gradient Sign Method* (FGSM) of adversarial attack . Perturbations are generated as $P = \epsilon \times sign\left(\mathcal{L}(x, y)\right)$, where $\mathcal{L}(x, y)$ is the model loss for datapoint $x$ and correct class $y$; and $\epsilon$ is the magnitude of the adversarial attack. We compare all models trained on MNIST and CIFAR-10 against adversarial FGSM attacks of different magnitudes.

Fig 5 shows the degradation in accuracy as the attack is intensified. For both datasets, CE performs consistently poorly, while VC shows consistent increased robustness to the adversarial attacks.

### 5.4 LOW DATA REGIME

Lastly, we briefly investigate model performance when data is scarce on the hypothesis that imposing a prior over the latent space may enable a model to generalise better from fewer samples. Each model is trained on only 500 training samples from the MNIST dataset.

| CE | GM | VC |
|---|---|---|
| $93.1 \pm 0.5$ | $\mathbf{94.4 \pm 0.3}$ | $94.2 \pm 0.4$ |

Table 4: Classification accuracy with low data (mean and standard error over 5 runs)

The results in Table 4 confirm that the prior, present in both the GM and VC models, improves model accuracy in a low data regime.

## 6 CONCLUSION

We have presented Variational Classification (VC), a generalisation of the softmax classifier, mirroring the relationship between the variational auto-encoder and the deterministic auto-encoder. We show that the softmax classifier is a special case of the VC model under specific assumptions that are effectively taken for granted when using the softmax output layer. We present a training objective to train a VC analogous to the ELBO, together with an adversarial optimisation regime. A series of experiments show that with little computational overhead, a variational image classifier outperforms the standard softmax in several ways, in particular in terms of calibration, adversarial attacks and when data is scarce, without degrading and potentially even improving classification accuracy.

The VC model opens up several interesting future directions. For example, $q(z|x)$ might be modelled as a stochastic distribution rather than a delta function and be trained to reflect uncertainty in the latent variables. Also, having a prior over latent variables may enable semi-supervised learning similar to other methods that work by implicitly imposing a latent prior.

In terms of limitations, we have focused on a particular aspect of a softmax classifier, how inputs to the softmax layer can be manipulated to make better predictions. A key outstanding question is what the rest of the neural network $f_\omega$ does up to that point, or indeed, should do.

## REPRODUCIBILITY STATEMENT

For our experiments, we adapt our implementations for our *ResNet-50* and *WideResNet-28-10* models from a public GitHub repository[4]. We use their implementation in the basic setting which we keep consistent throughout the CE, GM, and VC training approaches. Our experiments with MNIST were performed using a 6-layer CNN. For our discriminator, we experimented between using a single- and 2-layer neural network. We treated this choice as a hyperparameter.

**We will release our code and models upon acceptance.**

## ETHICS STATEMENT

We foresee no major ethical considerations for this work. The datasets used in this work do not contain any sensitive information to the best of our knowledge.

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

## A  CALIBRATION METRICS

One way to measure if a model is calibrated is to compute the expected difference between the confidence and expected accuracy of a model.

$$\mathbb{E}_{P(\hat{a}|q)}\Big[\mathbb{P}(\hat{a} = a|P(\hat{a}|q) = p) - p\Big] \tag{10}$$

This is known as expected calibration error (ECE) (Naeini et al., 2015). Practically, ECE is estimated by sorting the predictions by their confidence scores, partitioning the predictions in $M$ equally spaced bins $(B_1 \ldots B_M)$ and taking the weighted average of the difference between the average accuracy and average confidence of the bins. In our experiments we use 20 equally spaced bins.

$$\text{ECE} = \sum_{m=1}^{M} \frac{|B_m|}{n} |acc(B_m) - conf(B_m)| \tag{11}$$

### A.0.1  RELIABILITY DIAGRAMS

Another common tool to visualize model calibration is a reliability diagram. A reliability diagram plots sample accuracy as a function of confidence for each bin. If a model is perfectly calibrated, the confidence and accuracy bars should be identical.

## B  OOD DETECTION

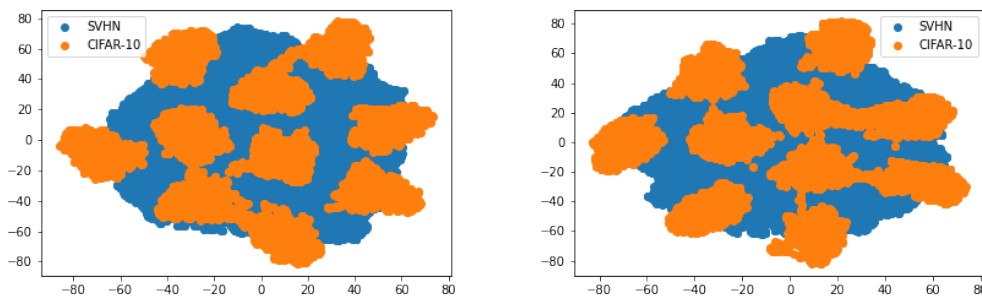

Figure 6: t-SNE plots of the feature space for a classifier trained on CIFAR-10. *(l)* Trained using CE. *(r)* Trained using VC. We posit that similar to CE, VC model is unable to meaningfully represent data from an entirely different distribution.

## C  CLASSIFICATION UNDER DOMAIN SHIFT

A comparison of accuracy between VC and CE under domain shift can be found in Figure 7. We find that VC performs comparably well as CE.

## D  SEMANTICS OF THE LATENT SPACE

In an effort to try to understand what semantics can the latent z space capture, we use a pre-trained MNIST model on the Ambiguous MNIST dataset (Mukhoti et al., 2021). We then interpolate the ambiguous 7's that spill close to the 1's and 2's Gaussian. We can qualitatively see that as we traverse from the mean of the 7's Gaussian to the 1's, the ambiguous 7's start to look more like 1's. Our latent space representation can be seen in Fig 8

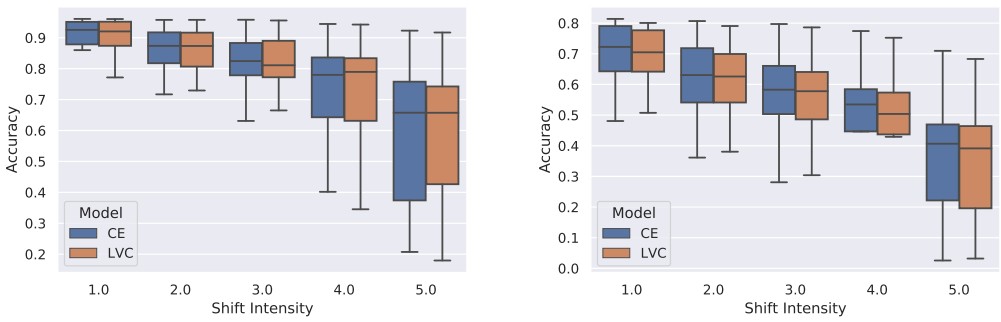

Figure 7: Classification accuracy under distributional shift, *(l)* CIFAR-10-C *(r)* CIFAR-100-C

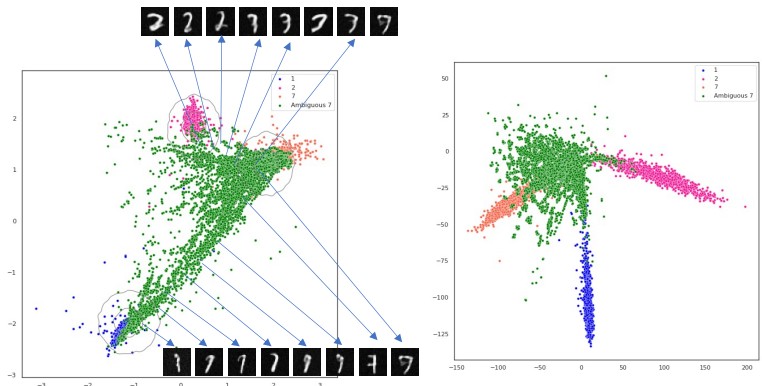

Figure 8: Interpolating in the latent space: Ambiguous MNIST when mapped on the latent space. *(l)* VC, *(r)* CE

