# OpenReview forum: "Variational Classification"
_ICLR.cc/2023/Conference — Submitted to ICLR 2023_

### Official Review · Reviewer_aooA · 2022-10-25

**Confidence:** 4
**Correctness:** 3
**Technical Novelty And Significance:** 2
**Empirical Novelty And Significance:** 2
**Recommendation:** 5

**Clarity, Quality, Novelty And Reproducibility:**

The idea to introduce the latent variable to classification task is interesting, but I think it will be better to add more theoretical analysis and experiments to support the claims.

**Strength And Weaknesses:**

Quality/Clarity: the paper is well written and the techniques presented are ok to follow. Its motivation is clear, which is to generalize softmax and design a better model: (1) modeling p(y|x) (2) measure of
similarity between data samples. On a technical level, I do not see how VC objective in Eq. 6 related to ELBO. The authors need to give more details derivation from Eq.5 to Eq. 6.

Originality/significance: the idea is interesting, which introduces latent variable and adds more constraints to generalize the softmax function. However, it gives readers an impression that the VC objective in Eq. 6 is a heuristic result, not from ELBO. If the authors can provide the detailed derivation to bridge the gap as well as show more experimental results, it should be an accepted paper.

**Summary Of The Paper:**

This paper presents a Variational Classification (VC), which generalizes the softmax classifier, and the authors claim that it mirrors the relationship between the variational auto-encoder and the deterministic auto-encoder. The main contribution is to include the latent variable to softmax classifier and then VC objective analogous to the ELBO is designed. The experimental results show that the VC classifier outperforms the standard softmax in several ways, in particular in terms of calibration, adversarial attacks and when data is scarce.


**Summary Of The Review:**

The paper adds a latent variable to classifier and impose constraints (class conditional priors) in the objective function.
Overall it is a good paper, but not ready for publication.

---

> ### Author Response · Authors · 2022-11-19
> **Thanks for the review, we've added the derivations along with more experiments**
>
> Thank you for your review. Please find detailed responses to your comments below.
> 1. **Derivation of Eqs 5 & 6**: please see the response to all reviewers regarding the derivation of the VC objective.
> 2. **VC objective seems heuristic**: we hope that improvements to the paper's clarity and detailed steps in Appendix 1 address this concern. To clarify, the VC objective is derived in logical steps by direct analogy to the ELBO. Although differences between the models being learned (*p(y|x)* vs *p(x)*) cause significant differences to the ELBO, strong similarilties remain: the nature of the three terms ("reconstruction", "prior", entropy"), that modelled distributions must be "consistent" with each other under Bayes' rule, and the KL divergence interpretation.
> 3. **More experimental results**: please see response to all reviewers regarding additional experiments.

---

### Official Review · Reviewer_qvVd · 2022-10-28

**Confidence:** 3
**Correctness:** 3
**Technical Novelty And Significance:** 2
**Empirical Novelty And Significance:** 3
**Recommendation:** 5

**Clarity, Quality, Novelty And Reproducibility:**

This paper introduces a lot of concepts, but in a relatively clear way. The quality of the paper can be improved by providing more experimental results. The paper does bring some novelty in addressing classification from a variational direction. The code is not provided and it seems non-trivial to implement the whole algorithm.

**Strength And Weaknesses:**

Pros:
1. Compared to deterministic methods, probabilistic ones like variational model can also smooth the predictions and alleviate overfitting issues. That's why this method can potentially help with calibration, interpolation, OOD, etc.
2. The paper explains the details very thoroughly, and carefully compare the difference and similarity between existing and proposed methods.
3. The experiments explored multiple domains/tasks.

Cons:
1. From a machine learning perspective, I think another view of understanding this paper is the embedding learning. Both input feature x and label y are mapped to a latent embedding z. The label learnt probabilistic space is the prior space and the input learnt probabilistic space is the posterior space. You kinda try to match the two spaces. While probabilistic method often brings smoothness, the performance loss is also concerned. In general, I feel this work still doesn't jump out of this framework and maybe suffer from the same issues?
2. Though this type of variational learning is relatively novel, the use of VAE in classification has been quite ubiquitous. The latent variable models also learn and align latent subspaces [1, 2]. I think you could discuss these works.
3. While this paper has experiments on multiple setups/scenarios, they look a bit thin to me. First, the major datasets are cifar-10, cifar-100, etc. In the computer vision domain, these are just too small. It's less convincing to only use these datasets. Also, you only tested WideResNet-28-10 and ResNet-50. More and larger models would give readers a more comprehensive view on the performance.
4. You can give some derivation steps for Eq. 5 and 6, at least in Appendix.
5. Figure 2: Variation Classier -> Variational Classifier

[1] Bai, J., Kong, S. and Gomes, C.P., 2022, June. Gaussian mixture variational autoencoder with contrastive learning for multi-label classification. In ICML (pp. 1383-1398). PMLR.

[2] Bai, J., Kong, S. and Gomes, C., 2021, January. Disentangled variational autoencoder based multi-label classification with covariance-aware multivariate probit model. In IJCAI (pp. 4313-4321).

**Summary Of The Paper:**

This paper proposes a variational classifier (VC). Typical machine learning classifiers use sigmoid or softmax to deterministically map last layer feature vector to class label predictions. This paper revisits the MLE, MAP and Bayesian under a variational framework. This work designs a novel objective based on ELBO and adversarial/contrastive technique. Experiments show that VC has some desired properties in interpolation, prediction confidence, out-of-sample detection, etc.

**Summary Of The Review:**

This paper is a more theory-oriented paper with empirical experiments support. The idea is interesting, though resembling several prior works. This paper should better shape its relations with literature and elaborate on the model details and experiments.

---

> ### Author Response · Authors · 2022-11-19
> **Thanks for you comments, we've added our derivations and conducted more experiments**
>
> Thank you for your review. Please find detailed responses to your comments below.
> 1. **"Embedding learning"**: We are not fully clear on this point, but hope that it is addressed by improvements to the paper's clarity, e.g. the addition of Algorithm 1 outlining the optimisation process, and Appendix 1 which more clearly steps through the derivation of the VC objective (please see response to all reviewers regarding the derivation of the VC Objective). Please expand on this point further if it is not fully addressed.
> 2. **Related Works**: we have added references (§4, last paragraph) to a number of works (including your reference [2]) that impose a mixture of Gaussians distribution in the latent space.
>     - Please note that we draw several analogies to the VAE, and we proposed a variational method to learn model parameters, but our model is not at all generative and is not directly comparable to generative approaches, e.g. VAEs.
>     - Although it is not fully distinct, we do not see a clear connection to your reference [1], which combines multiple loss function components, and involves cumulative density functions and MCMC.
> 3. **More experimental results**: please see response to all reviewers regarding additional experiments.
> 4. **Derivation of Eqs 5 & 6**: please see response to all reviewers regarding derivation of VC objective.
> 5. **Figure 2**: Thanks - typo fixed.
> 6. **Code provision**: We will make the code publicly available at the camera-ready stage, if accepted.

---

### Official Review · Reviewer_Lvkv · 2022-11-02

**Confidence:** 4
**Correctness:** 3
**Technical Novelty And Significance:** 3
**Empirical Novelty And Significance:** 2
**Recommendation:** 5

**Clarity, Quality, Novelty And Reproducibility:**

The paper is clear. For reproducibility, I think more details should be provided.

**Strength And Weaknesses:**

Strengths:

- The generalization of the softmax classifier with a Bayesian interpretation is interesting and it can potentially have impacts on the several desired properties for classification models such as calibration and OOD generalization.

- The authors provide a derivation for the VC objective with nice connections to VAE. I also liked that they provided practical implementation points of the VC using the commonly employed tools like the reparameterization trick. However, I believe it could be useful to provide an algorithm table or pseudo-code for better clarity. Could the authors also clarify if there are any instability issues faced during the adversarial optimization?

- The experiments were selected to validate different aspects of the proposed VC. More on that below.

Weaknesses: My main concerns are about the experimental results.

- From the results, it is hard to say VC is doing notably better than GM or CE methods, except model calibration (even for that, it is sometimes worse than CE, see Fig. 4. left). Could this be due to optimization difficulties, e.g. due to adversarial objective, or are there other reasons?

- For calibration, adding temperature scaling as another baseline could be helpful to gauge the improvements.

- Can this method be applicable to more complex tasks like ImageNet classification and more strong adversarial attacks like PGD? Also for low data regime, are there any benefits for datasets other than MNIST where the performance is more or less saturated? What happens to calibration under low data regime?

- Demonstration of the experimental results are somewhat inconsistent. For example, why is there no standard deviation in Table 1 from multiple runs while they are provided for Table 4? What do the bars and whiskers represent in Fig. 4?

- Is there a substantial increase in the training time compared to standard training? An analysis on this could be useful for practitioners.

- Outlining the experimental protocol, e.g. used augmentations and hyperparameters, could increase the reproducibility of the work.

- I found Sec. 3.3 to be somewhat confusing. Did the authors observe whether the model learned any disentangled and semantically meaningful representations?

Minor: There are several typos throughout the paper. Also, the font size in figures are too small.


**Summary Of The Paper:**

This paper introduces variational classification (VC) by treating the inputs to the softmax layer of a classification model as latent variables with a mixture of Gaussians prior. To achieve this probabilistic interpretation of the softmax classifier, they derived an objective to be minimized, which is similar to the ELBO used in VAEs. The resulting VC model generalizes the softmax classifier (similar to VAE vs deterministic autoencoder) and enables incorporating class-conditional priors. A derivation of the training objective is provided with some practical design choices to optimize it. The evaluations are performed on several datasets (CIFAR-10, CIFAR-100, SVHN, CelebA, MNIST) in terms of accuracy, calibration, OOD generalization, adversarial robustness, and performance on low data regime.

**Summary Of The Review:**

While the paper makes some interesting points, I think it is still a borderline one as the performance is somewhat mixed and gains are mostly incremental. I would be happy to revise my score if the above concerns are addressed sufficiently.

---

> ### Author Response · Authors · 2022-11-19
> **Thanks for your comments, we've improved the explanation and conducted more experiments**
>
> Thank you for your review. Please find detailed responses to your comments below.
>
> 1. **Pseudo-code for training the VC**: this has been included as Algorithm 1 (p.6)
> 2. **Training instability**: we confirm that no instability was found during the adversarial training. The loss and accuracy curves progress smoothly.
>     * Since these plots show typical behaviour we do not include them in the paper but to evidence this we post plots of (i) training loss; (ii) prediction accuracy of the auxiliary binary classifier for $z \sim q(z|y)$ ("real") and (iii) $z \sim p(z|y)$ ("sampled") latents; and (iv) prediction accuracy on the development set here: https://imgur.com/a/Rs8UaJl (anonymised link).
>     * Note that the adversarial nature of a VC is quite the opposite of a GAN: in GANs a Gaussian is mapped (through a NN) to a complex distribution that the classifier must distinguish from a highly complex natural distribution; in a VC a potentially complex distribution is mapped to become a simple Gaussian and distinguished from a Gaussian. We include a discussion on this in §3 (final paragraph).
>     * We note also that previous work suggests that GAN training instability may relate to the target distribution (p(x)) lying on a low-dimensional manifold in a high dimensional space ["Wasserstein GAN", Arjovsky et al. (2017)], whereas our target distribution is a simple Gaussian in the relatively low dimensional latent space.
>
> 3. **VC vs GM**: please see response to all reviewers regarding VC vs GM.
>     * please note that the differences in Fig 4 (left) are not statistically significant (see error bars). Fig 4 (right) shows statistically significant differences for a more complex dataset. We confirm that no optimisation difficulties are encountered due to the adversarial objective (or otherwise).
> 4. **Temperature Scaling**: we have added results for this in Fig 4.  (see response to all reviewers regarding additional experiments).
> 5. **ImageNet/PGD**: please see response to all reviewers regarding additional experiments.
> 6. **Consistent presentation of results**: standard deviations are included for results in tables 1, 2 and 4.
>     * bars and whiskers in Fig 4: Details have been added to the figure caption.
> 7. **Training time increase/analysis**: The VC model causes no significant increase in training time. The wall clock time to train a wide-resnet-28-10 for 200 epochs increased by ~15 mins for VC relative to CE (c 5.5 hours).
> 8. **Experimental protocol**: We include many implementation details in the paper, but will also release our code (if accepted) for full transparency and reproduciblity.
> 9. **Section 3.3**: This section considered potential justifications for assuming a mixture of Gaussian latent distribution, e.g. matching the generative process or by a central limit theorem argument. This is not a core part of the paper and has been moved to Appendix B to allow for Algorithm 1 which gives pseudo-code for the VC training process.
>     * We agree that properties of learned representations are of interest and plan to investigate this in future work.
> 10. **Typos/Font size**: We have corrected and typos found and will thoroughly proof read again at the camera-ready stage, if accepted. Fonts in figures have been improved.
> 11. **Incremental gains**: We have added more experiments, including repeated runs of previous experiments, to demonstrate that performance improvements are statistically significant and strengthen evidence for the hypothesis that a latent variable approach to classification (effectively regularising and constraining the latent space) improves classification robustness.

---

### Author Response · Authors · 2022-11-19
**Response to all reviewers**


We thank all reviewers for your time and feedback on our work. We are very pleased that all reviews find positive aspects to the paper. We address here any points raised by multiple reviewers and separately provide detailed responses to each reviewer.

1. **Clearer derivation of VC Objective (Eqs 5 & 6)**, :
Thank you for your feedback on this important aspect of the paper, we recognise that the derivation for the VC objective and motivation for the steps involved was insufficiently clear. We have revised wording in the main body of the paper to address this and included a fuller step-through of the derivation in Appendix 1. We also include pseudo-code for training a VC in Algorithm 1, p.6 (as suggested by Reviewer Lvkv)). To clarify, whilst the VC objective is guided in the direction of generalising softmax cross-entropy, each step should be logical in its own right, with any implications well justified. We very much hope this is now easier to follow and the steps are more clear but welcome any further feedback.

2. **Additional experiments**:
As requested by all reviewers, we have included a number of additional experimental results, which confirm our previous findings:
   * Low data regime (§5.4): We include classification experiments in a low data setting for CIFAR10 (previously only MNIST), showing that training objectives with a prior (VC & GM) consistently outperform cross entropy (CE) when training on fewer samples. Further, VC outperformance GM on the more complex CIFAR10 dataset.
   * Larger data sets: We are running classification experiments on larger data sets (e.g. Imagenet) and plan to include their results in the paper.
   * Repeated runs (Tables 1, 2 & 4): For consistency and robustness, we report results for prediction accuracy and calibration scores showing mean and standard error over 5 model runs (random seeds).
   * Calibration with temperature scaling (Fig. 4): we include results of temperature scaling on the OOD calibration task, demonstrating that VC is more calibrated on OOD data.

3. **VC vs GM vs CE**:
As a point of clarification, results for the "GM" model (softmax with mix-of-Gaussian latent prior) are included for several tasks as an ablation between the softmax cross-entropy (CE) baseline and the variational classifier (VC). This is to help understand which component of the VC objective leads to performance improvement. For example, if the VC and GM models comparably outperform CE, it suggests that the prior term is beneficial for that task, whereas entropy ("puffing out" class-conditional latent distributions) is less relevant. For example, intuitively, entropy seems less important for making accurate classifications than for assessing the confidence of those predictions, which is shown between Tables 1 & 2.

---

### Decision · Program_Chairs · 2023-01-20

**Decision:**

Reject

**Justification For Why Not Higher Score:**

The idea and the theory of the paper is valid. However, it is not at the state to be published. It needs a significant improvement on the empirical evaluation side. Moreover, it is also not possible to see the state of the writing of the paper since authors missed the deadline to upload their revision.

**Justification For Why Not Lower Score:**

N/A

**Metareview: Summary, Strengths And Weaknesses:**

The paper is proposing a replacement of final linear classification of a deep network with a probabilistic classifier using variational learning. Since the classifier has a clearer probabilistic meaning in the proposed setup, it is expected to perform better when the probabilistic meaning is critical. Problems like OOD detection and handling adversarial inputs are all invalidating iid. assumption and requiring reliance of some form of a prior which this method can provide and/or utilise.

Strengths:
- A clear and theoretically sound idea. The overall promise of the variational final layer is sensible and follows from first principles.
- A flexible approach. The proposed approach is flexible to include other type of priors and domain knowledge which makes it a good candidate to handle OOD family of problems.
- Promising initial experiments. Authors provide some promising results.

Weaknesses:
- The paper was not clear enough for the reviewers. All reviewers shared the concern that some crucial knowledge is missing. After my personal reading, I agree that the paper is not self contained as some derivations are pushed out. Although authors tried to fix this during their rebuttal, they missed the deadline due to partly a technical issue and partly because they failed to upload the new version in a timely manner. Authors added them as a comment but I believe they should be part of the main text not appendix, at least partially. It is not possible to review the updated paper without seeing it since I am not sure how would it fit to the page limit and what kind of changes authors would make. This is a major issue.
- The experiments are promising but all comparisons are self baselines. Authors need to show that their approach is useful for some existing applications compared with some existing baselines. Current results and experiments are clearly enough to validate the idea and its soundness. However, it is not enough to validate its applicability and usefulness. This issue also persists through the fact that all experiments are on small datasets and some specific model families. To show the usefulness, I strongly recommend authors to add their method on top of an existing method which solves an existing benchmark and show its value in such a realistic setup.